Original research

# Organisation of care for people receiving drug-resistant tuberculosis treatment in South Africa: a mixed methods study

Lindy Dickson,[1] Sacha Roxanne Le Roux ,[1] Leila Mitrani,[1] Jeremy Hill,[2] Waasila Jassat,[3] Helen Cox ,[1,4] Koleka Mlisana,[5] John Black,[6] Marian Loveday ,[7] Alison Grant,[2,8] Karina Kielmann,[9,10] Norbert Ndjeka ,[11,12] Mosa Moshabela,[12] Mark Nicol [1,13]

For numbered affiliations see end of article.

**Correspondence to**
Professor Mark Nicol;
Mark.Nicol@uwa.edu.au

## ABSTRACT

**Objectives** Treatment for multidrug-resistant/rifampicin-resistant tuberculosis (MDR/RR-TB) is increasingly transitioning from hospital-centred to community-based care. A national policy for decentralised programmatic MDR/RR-TB care was adopted in South Africa in 2011. We explored variations in the implementation of care models in response to this change in policy, and the implications of these variations for people affected by MDR/RR-TB.

**Design** A mixed methods study was done of patient movements between healthcare facilities, reconstructed from laboratory records. Facility visits and staff interviews were used to determine reasons for movements.

**Participants and setting** People identified with MDR/RR-TB from 13 high-burden districts within South Africa.

**Outcome measures** Geospatial movement patterns were used to identify organisational models. Reasons for patient movement and implications of different organisational models for people affected by MDR/RR-TB and the health system were determined.

**Results** Among 191 participants, six dominant geospatial movement patterns were identified, which varied in average hospital stay (0–281 days), average patient distance travelled (12–198 km) and number of health facilities involved in care (1–5 facilities). More centralised models were associated with longer delays to treatment initiation and lengthy hospitalisation. Decentralised models facilitated family-centred care and were associated with reduced time to treatment and hospitalisation duration. Responsiveness to the needs of people affected by MDR/RR-TB and health system constraints was achieved through implementation of flexible models, or the implementation of multiple models in a district.

**Conclusions** Understanding how models for organising care have evolved may assist policy implementers to tailor implementation to promote particular patterns of care organisation or encourage flexibility, based on patient needs and local health system resources. Our approach can contribute towards the development of a health systems typology for understanding how policy-driven models of service delivery are implemented in the context of variable resources.

## STRENGTHS AND LIMITATIONS OF THIS STUDY

⇒ We used real-world patient movement pathways to understand emerging models of care in response to a change in policy.
⇒ We triangulated several different sources of data, including laboratory data, interviews with healthcare providers and facility visits to understand the implications of these emerging models of care for people affected by multidrug-resistant/rifampicin-resistant tuberculosis (MDR/RR-TB) and the health system.
⇒ We could not clearly differentiate to what degree patient needs vs health system organisation and resources determined individual care pathways.
⇒ We were not able to adequately assess care quality indicators, and so are unable to determine whether particular organisational models were associated with favourable patient outcomes.
⇒ We did not interview people affected by MDR/RR-TB, and so cannot draw conclusions about their preferences.

## BACKGROUND

Tuberculosis (TB) has been the leading cause of death in South Africa for two decades[1] and claims the lives of around 330 persons daily worldwide.[2] Multidrug-resistant (MDR-) TB is defined as tuberculosis caused by *Mycobacterium tuberculosis* that is resistant to at least the two key drugs, rifampicin and isoniazid, while rifampicin-resistant (RR-) TB is defined by resistance to rifampicin with or without isoniazid resistance. MDR/RR-TB caused illness in around 14 000 persons in South Africa in 2019,[3] is more complex and costly to treat than drug sensitive TB,[4] and is associated with a fourfold higher mortality.[1 5] Within the next 20 years, the MDR/RR-TB case load in South Africa is projected to increase substantially.[6 7] Persons with MDR/RR-TB often experience complex medical, psychological,[8]

social[9] [10] and economic[11] challenges. Management for people with MDR/RR-TB requires an intensive multi-disciplinary, comprehensive approach and typically consumes a disproportionate share of health resources.[11] Treatment provision requires access to specialised drugs and skilled staff to monitor people with MDR/RR-TB over many months.[11] The sustainable use of newer regimens to address this case load will require strengthening MDR/RR-TB treatment delivery systems to ensure effective, person-centred care.[12] The WHO End TB Strategy advises a country-specific, tailored and strategic approach to achieving universal access to high-quality treatment for MDR/RR-TB.[13] [14]

In response to the rising MDR/RR-TB caseload in South Africa and to local and international pressure from patient and advocacy groups, the National Department of Health adopted an approach to increase ambulatory management of MDR/RR-TB in 2011.[15] This strategy, which had not been widely implemented elsewhere, but was in line with pilot projects in South Africa, Peru and Lesotho, was intended to improve treatment access and reduce in-hospital stays.[16] [17] Prior to decentralisation, people with MDR/RR-TB in South Africa were primarily managed within facilities specialising in MDR/RR-TB care, frequently with long hospital stays, before being discharged under the ongoing care of the specialised facilities. The decentralisation strategy aimed to increase the number of treatment sites from one or two dedicated MDR/RR-TB facilities per province to several facilities capable of providing MDR/RR-TB care within each district.[18] A national guideline for decentralisation of care was published in 2011[19] and updated in 2019,[20] and served as a generic, non-prescriptive tool for health managers. Since then, implementation of the policy to decentralise MDR/RR-TB care within South Africa has varied widely,[21] in response to differences in health system capacity and readiness to cope with change.[22] Funding to support decentralisation was not specifically allocated, which resulted in further setting-specific innovations and adaptations to the programme at both provincial and district level.[23] [24]

We have previously described variation in the care journeys of people with MDR/RR-TB,[21] which may reflect differences in patient comorbidity or disease features, but likely also reflects regional differences in the organisation of care for the management of MDR/RR-TB. Ideally, models for organisation of care should attempt to improve patient outcomes through coordinated care that is person-centred, timely and effective[25]; however, models of care often evolve to suit healthcare providers and health systems convenience.[26–28]

We therefore studied variation in the patterns of organisation of care for people with MDR/RR-TB which have emerged in South Africa because of local variation in guideline implementation. We used patient care pathways to identify patterns of care provision, outline the implications of these patterns for the needs of people with MDR/RR-TB and draw lessons on different ways

in which care for patients with complex illness can be organised.

## METHODS

The overarching research project, within which this substudy was nested, was led by the same group of investigators, funded by a Health Systems Research Initiative award from the Medical Research Council of the UK and aimed to identify interventions to optimise decentralisation of services for people with MDR/RR-TB (online supplemental figure 1). The project used a stepwise realist approach to understand the policy context, implementation and working models of decentralisation of MDR/RR-TB care in South Africa. The aim of this substudy, which was conducted over the same period, was to explore variations in the implementation of care models in response to a change in MDR/RR-TB health policy, and the implications of these variations for people with MDR/RR-TB and the health services.

### Study design and setting

We used a multiple case study methodology.[29] Data collection included a retrospective cohort study of geographically-linked laboratory records, validated by patient folder reviews. All TB samples in the state sector in South Africa are sent to laboratories which form part of the National Health Laboratory Services (NHLS) for testing. The NHLS has a centralised data repository which records all samples sent to laboratories within the network, together with results and patient details. Patterns of patient movement and facility visits were mapped. Semi-structured interviews with healthcare workers were undertaken to verify patient journey patterns and organisational patterns, and to determine their implications for the health service and people with MDR/RR-TB. We defined MDR/RR-TB as resistance to rifampicin (with or without isoniazid resistance), detected using genotypic or phenotypic methods, irrespective of resistance to other anti-TB drugs.

Three provinces in South Africa (KwaZulu-Natal (KZN), Eastern Cape (EC) and Western Cape (WC)) were selected due to their high MDR/RR-TB disease burden and differing approaches to decentralised care (online supplemental figure 2). Within each province, two urban and two rural districts were selected based on observed differences in patient pathways from the parent study.[30] One additional rural district was selected in KZN, due to a rapid shift in urbanisation associated with relocation of the province's airport.

The study took place between July 2016 and July 2019. During this period, MDR/RR-TB treatment regimens were largely based on a standardised aminoglycoside-containing regimen (requiring an injectable drug), tailored according to individualised drug-susceptibility patterns. Prior to the study period, rapid molecular testing for TB and MDR/RR-TB (Xpert MTB/RIF) had been implemented, while limited roll-out of the novel

anti-tuberculous drug bedaquiline in pilot programmes and the introduction of a shortened MDR/RR-TB treatment regimen occurred over the study period. People with MDR/RR-TB generally underwent scheduled monthly clinical evaluation and investigation over the treatment duration, in line with WHO guidelines.[30]

### Selection of people with MDR/RR-TB and assessment of patient movement pathways

We have previously described the methods used to construct MDR/RR-TB patient healthcare journeys.[30] Briefly, NHLS laboratory records were used to generate a list of all people in the study districts with new bacteriologically-confirmed MDR/RR-TB from July to September 2016 (n=2649). A random sample of 195 people with MDR/RR-TB was selected from this list, stratified by district. One patient was found to have been incorrectly diagnosed with MDR/RR-TB, leaving 194 in the analysis. The laboratory records for the 194 people served as tracers of their pathways over a 9-month period.[30] Geographical coordinates of healthcare facilities were collected, and spatial-temporal mapping of patient movement was used to reconstruct patient movement pathways in chronological order for each sampled person with MDR/RR-TB, including movement outside the study districts.

Data on treatment progress were collected from clinical record reviews, telephonic enquiries with clinics and the electronic MDR/RR-TB registry (EDRWeb) up to 2 years after diagnosis. Data on patient movement obtained from laboratory records were confirmed by telephonic enquiries at all healthcare facilities attended by people with MDR/RR-TB. This input was used to verify laboratory information, determine additional clinical visits (where no laboratory sample was taken), obtain demographic and health data and to identify reasons for movement of sampled people with MDR/RR-TB between facilities. Descriptive statistics for the sampled districts and participants were extracted. Time from laboratory diagnosis to treatment initiation and time on treatment were calculated.[31]

All movement between healthcare facilities for each study participant was geospatially mapped, based on 'as the crow flies' measurements.[32] Patient pathway patterns sharing similar appearance were then grouped according to their pattern and named, as described in other studies.[33–35] Pattern descriptions included the level of facilities visited (ranging from quaternary care level to mobile clinic), movements between facilities, facility admission period or 'period under care' (where facilities retained responsibility for monthly follow-up of people with MDR/RR-TB after discharge) and sector of care (state or private). The number of visits (or patient-days) spent under care of a centralised service were used to position the patterns on a decentralisation continuum.

### Detailed assessment of organisation of services for MDR/RR-TB

A total of 22 facilities representing the spectrum of patterns were chosen for more detailed investigation of organisation of MDR/RR-TB care. These facilities represented six urban and seven rural districts as well as the different levels of care ranging from primary healthcare to tertiary care. At each facility we conducted patient folder reviews and semi-structured interviews with health workers, including pharmacists (n=2), outpatient physicians (n=2), inpatient physicians (n=6), TB facility managers (n=5), ward nurses (n=9), outpatient nurses (n=8) (including HIV care nurses (n=2), paediatric care nurses (n=2)) and lay healthcare workers (n=2). Healthcare workers were selected for interview based on their involvement with the MDR/RR-TB care programme at their facility. No participants refused to participate or dropped out and only the researchers and participant/s were present during interviews. For interviews, specific patient trajectories related to each facility were used as prompts for discussion of the reasons for and implications of patient movements between facilities. Additional data on the same themes were extracted from patient folder reviews. The interviews were conducted in English, with some translation from Afrikaans, and recorded, except in a prison where recording was not permitted. Most interviews were conducted with individuals or pairs of informants, with two interviews involving multidisciplinary panels at a facility. Transcripts of interviews and folder review data were reviewed by seven researchers who extracted information relating to the implications of the different patterns for care organisation for both people with MDR/RR-TB and the health services. A Consolidated criteria for Reporting Qualitative research checklist is included (online supplemental material). To further identify the implications of the different organisational patterns for care outcomes and person-centredness for people with MDR/RR-TB and the health services, we used key metrics including time to treatment, length of hospital stay, number of facilities visited and distance travelled.

### Patient and public involvement statement

This study involved neither patients nor members of the public. Findings from this study were disseminated during national MDR/RR-TB meetings organised by the Department of Health and non-governmental organisation partners and presented to programme implementers from all nine provinces of the country.

## RESULTS
### Cohort description

We studied urban and rural districts from three high TB burden provinces (online supplemental table S1). A total of 191 participants had sufficient data available to map treatment pathways and outcomes (the remaining three participants had too few data points to assess).[36] Of the

191 people with MDR/RR-TB, 63% (n=120) were men and 92% (n=177) were aged 20–59 years. Approximately half of the participants lived in an urban area (46%, n=88). Sixty-three per cent (n=121) had previously been treated for TB, with a further 19% (n=36) previously treated for MDR/RR-TB. Sixty-nine per cent (n=131) were coinfected with HIV at the time of MDR/RR-TB diagnosis.[36] Two-thirds (65%, n=124) were unemployed at the time of diagnosis. Fifteen per cent (n=28) had a history of correctional services detention. Approximately one quarter (24%, n=46) reported comorbidities other than TB and HIV at diagnosis. Five per cent (n=10) had a household member die from TB disease. All participants were treated exclusively in the state health sector, except for two participants who also received treatment in the private sector. We have previously described in detail the clinical comorbidities and social needs of the people with MDR/RR-TB in this cohort.[36]

Sixty-four per cent (n=122) of the cohort were diagnosed with MDR/RR-TB; 16% (n=31) with pre-extensively drug-resistant (pre-XDR), 10% (n=19) with XDR-TB and 10% (n=19) with MDR/RR-resistance with unknown second-line drug resistance. Of the 191 participants 80% (n=153) were alive at 6 months, and 77% (n=148) were alive at 2 years after laboratory diagnosis. At 24 months after laboratory diagnosis, 46% of participants (n=88/191) had either completed treatment or achieved 'cure', 19% (n=37) had ongoing treatment, 10% (n=19) were lost to follow-up, 2% (n=3) had unknown outcome and 0.5% (n=1) had ongoing illness where TB treatment was deemed futile and the patient had been referred for palliative care.

### Geospatial patterns of patient movement and their distribution

A total of 191 patient movement pathways could be classified into six dominant geospatial movement patterns (table 1 and figure 1). These were named according to the trajectories of patient movements as A: 'fan' (n=41 (22%)), B: 'chicken-foot' (n=19 (10%)), C: 'hub and spoke' (n=72 (38%)), D: 'triangle' (n=30 (15%)), E: 'line' (n=12 (6%)) and F: 'dot' (n=17 (9%)). The patterns were placed on a continuum from the most centralised, 'fan', to the most decentralised, 'dot'. The 'hub and spoke' configuration was the most frequently observed pattern across all three provinces in both urban (n=35 (18%)) and rural (37 (19%)) settings (table 2). Several typical patient pathways for each pattern are depicted on a stylised study district in figure 1.

### Description of organisation of care within each of the patterns

We used facility visits, staff interviews and patient record review to describe the organisation of care within each of these patterns. The most centralised pattern, 'fan', is highly dependent on the designated MDR-TB Centre of Excellence (COE) to manage people with MDR/RR-TB, following initial assessment at a peripheral facility. COEs were historically designated by the provincial health departments to manage people with complex TB. The

first step in increasing decentralisation of MDR/RR-TB services was the establishment of specialised MDR-TB hospitals, as a 'step down' from the COEs. The 'chicken-foot' pattern demonstrates some decentralisation to these specialised hospitals for routine cases, but with dependence on the COEs for caring for people with more extensive TB drug resistance or complex presentations. This pattern was common for regions with higher XDR-TB or MDR/RR-TB numbers and where new drugs were being introduced into TB regimens. Although the participant numbers in the study limited outcome analysis, observation suggests that this model has the greatest risk for people with MDR/RR-TB for retention in care, treatment outcome and prolonged hospital stay (table 1). The 'hub and spoke' pattern, which was predominately observed in rural districts, primarily uses specialised MDR-TB hospitals for all care provision, with little capacity existing at primary care level. These three models are considered the more centralised of the six models identified.

The 'triangle' pattern incorporates a more flexible network of MDR-TB services and indicates availability of MDR-TB services at many facilities within the district. Typically, this pattern has oversight by a specialist infectious diseases or TB physician, who is either within the district or closely monitors patient progress through telemedicine. This 'conductor' is responsible for weighing complex social and medical needs and for directing patient movement to an appropriate level of care, according to the clinical need and social circumstances of the patient, at various points during their care. This organisational model is also able to respond to changes in the local availability of MDR-TB resources, by modifying patient referral pathways. This model is frequently associated with decision-making shared between people with MDR/RR-TB and clinicians.

The 'line' and 'dot' patterns reflect a substantial shift to outpatient clinical-based care, frequently including MDR-TB clinical practitioners providing mobile outreach services to different subdistrict primary healthcare sites. In the line pattern, not all clinics had capacity to initiate treatment, which resulted in some clinics referring to neighbouring clinics or a district hospital for outpatient treatment initiation or a short admission for early management and stabilisation. In the 'dot' pattern, clinics had capacity to initiate and maintain MDR/RR-TB treatment independently, with pharmacy and multidisciplinary resources available. This 'dot' model was associated with the best treatment outcomes, however the sample size was small and possible healthcare worker selection of participants with less severe illness (to cope with a fully decentralised care option) limits generalisability of these findings.

Different patient movement pathways may reflect organisational models within the health system, the incidence of MDR/RR-TB within districts as well as patient features which require delivery of care at a particular level.[37] We observed that the distribution of the different movement patterns across districts differed. Most districts had three

**Table 1** Dominant geospatial movement patterns observed within study districts, together with key features of organisation of care within these patterns, descriptive statistics of patient journeys and treatment status after 24 months

| Pattern name | Fan | Chicken foot | Hub and spoke | Triangle | Line | Dot |
|---|---|---|---|---|---|---|
| Level of decentralisation | Highly centralised care. | Care decentralised to district level with high reliance on COE. | Care decentralised to district level. | Multilevel/ multifacility decentralised care. | Community based, ambulatory care. | Community based, ambulatory care. |
| Description of care delivery | Hospital-based care with long admissions, far from home. | Hospital-based care with long admissions, some care closer to home. | Hospital-based care with long admissions, closer to home. | Flexible care, coordinated by specialist doctors. | Home-based care, with short admissions if required. | Home based, outpatient care. |
| Diagnosis | At clinic or general¶ hospital. | At clinic or general¶ hospital. | At clinic or general hospital. | At clinic or general¶ hospital. | At clinic. | At clinic. |
| Treatment initiation | At provincial COE. | At specialist TB hospital, but with frequent change to regimen at COE. | At specialist TB hospital. | At specialist TB hospital or by specialist TB outreach service. | At MDR/RR-TB clinic or district hospital MDR/RR-TB clinic. | At the diagnosing clinic. |
| Coordination of care and follow-up | Specialist TB doctors at the COE. | Specialist TB doctors at the COE and TB doctors at the decentralised site. | TB doctors at the specialised site. | Specialist physician or TB physician on site or through telemedicine. | TB nurses at clinics. Sometimes assisted by a generalist doctor. | TB nurses at clinics. Sometimes assisted in person/telephonically by generalist doctor. |
| Hospitalisation | At COE, frequently for long periods. | At specialist TB hospital or COE, frequently for long periods. | At specialist TB hospital for long periods. | Depending on patient's medical and social circumstances. | Depending on patient's medical and social circumstances. | No admission. |
| Number of participants | N*=41 (urban n*=25, rural n=16) | N=19 (urban n=5, rural n=14) | N=72 (urban n=29, rural n=43) | N=30 (urban n=13, rural n=17) | N=12 (urban n=4, rural n=8) | N=17 (urban n=10, rural n=7) |
| Median distance from home to diagnosing site† | 10 km (n=36) | 11.4 km (n=17) | 14.6 km (n=66) | 10.6 km (n=24) | 5.4 km (n=6) | 0.9 km (n=14) |
| Median distance from home to initiating site† | 44 km (n=36) | 38 km (n=17) | 41 km (n=66) | 26 km (n=24) | 6 km (n=6) | 2 km (n=14) |
| Median distance travelled between facilities | 102 km | 188 km | 51 km | 76 km | 8 km | 0 km |
| Median time to treatment initiation‡ | 35 days | 13 days | 14 days | 7 days | 0 days | 1 day |
| Median (range) number of trips§ | 6 (1–7) | 5 (2–12) | 6 (1–17) | 8 (2–16) | 6 (1–21) | 8 (6–22) |

Continued

**Table 1** Continued

| Pattern name | Fan | Chicken foot | Hub and spoke | Triangle | Line | Dot |
|---|---|---|---|---|---|---|
| Median duration of hospital admission | 79 days (IQR=105) range 0–240 | 103 days (IQR=91) range 20–197 | 81 days (IQR=124 range 0–281 | 40 days (IQR=57) range 0–175 | 12 days (IQR=24) range 0–38 | 0 days (IQR=0) range:0–0 |
| Vital status 6 months following MDR/RR-TB laboratory result | Deceased 20%[8] | Decreased 11%[2] | Deceased 25% [18] | Deceased 20%[6] | Deceased 33%[4] | Deceased – nil |
| Treatment status at 24 months following MDR/RR-TB laboratory result | Cured/completed treatment 45%[19] Still on treatment 21%[9] (favourable 68%) Deceased 24%[10] Lost to follow-up 5%[2] Treatment failure nil Unknown 3%[1] | Cured/completed treatment 33%[6] Still on treatment 16.5%[3] (favourable 47%) Deceased 16.5%[3] Lost to follow-up 37%[7] Treatment failure nil. Unknown nil | Cured/completed treatment 41%[28] Still on treatment 21%[15] (favourable 61%) Deceased 26%[19] Lost to follow-up 9%[6] Treatment failure 2%[1] Unknown 3%[2] | Cured/completed treatment 53%[16] Still on treatment 17%[5] (favourable 70%) Deceased 20%[6] Lost to follow-up 10%[3] Treatment failure nil Unknown nil | Cured/completed treatment 17%[2] Still on treatment 42%[5] (favourable 58%) Deceased 42%[5] Lost to follow-up nil Treatment failure nil Unknown nil | Cured/completed treatment 94%[16] Still on treatment nil (favourable 94%) Deceased nil Lost to follow-up 6%[1] Treatment failure nil Unknown nil |

*N/n=number of participants.
†Distance travelled calculated as straight-line distance as calculated from the patient's home to the healthcare site.
‡Days from MDR/RR-TB diagnosis to treatment initiation.
§Number of trips undertaken to healthcare sites during the first 9 months of treatment.
¶General hospital indicates either a district or regional level hospital.
COE – TB, Centre of Excellence; MDR, multidrug-resistant ; RR, rifampicin-resistant ; TB, tuberculosis.

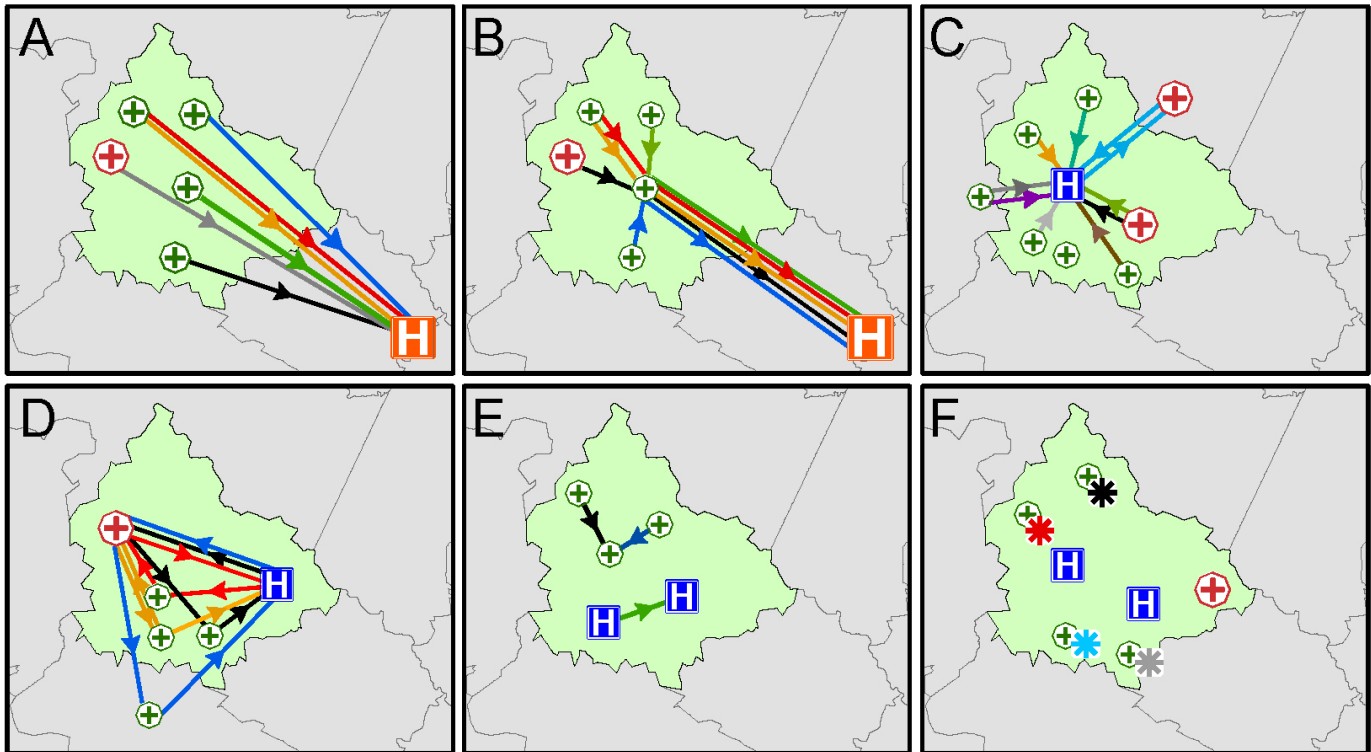

**Figure 1** Examples of the dominant geospatial movement patterns annotated on a stylised study district. Each coloured line depicts a single participant travelling in a pathway between the specific levels of healthcare facilities found in that model. Arrows represent direction of participant movement direction between healthcare facilities. Stars represent participants who only visited a single facility across their diagnostic and treatment pathway. Green crosses represent primary healthcare clinics, red crosses represent general hospitals, blue H represents specialised TB hospital and orange H represents Centre of Excellence TB hospital. A: 'Fan'; B: 'Chicken Foot'; C;' Hub and Spoke'; D: 'Triangle'; E: 'Line'; F: 'Dot'. TB, tuberculosis.

or four patterns identified (table 2). This suggests flexibility in the organisation of care within these districts to respond to differing patient or health system needs.

We conducted exploratory analysis to assess whether patient clinical features were associated with certain patterns of care (table 3). Children, participants older than 60 years and people with XDR-TB/ pre-XDR-TB seldom followed the more decentralised 'line' and 'dot' patterns. There was otherwise no evidence that patterns of care differed by patient HIV status, previous TB

**Table 2** The distribution of organisational models by study district

| Province | Rural or urban | Districts | A Fan | B Chicken foot | C Hub and spoke | D Triangle | E Line | F Dot |
|---|---|---|---|---|---|---|---|---|
| KwaZulu-Natal | Urban | Ethekwini | 4 | 3 | 5 | 1 | | 1 |
| | | Uthungulu | | 4 | 2 | 7 | | 2 |
| | Rural | Ilembe* | 2 | | 8 | 5 | | |
| | | Umkanyakude | | 2 | 7 | 6 | | |
| | | Uthukela | | 6 | 8 | 1 | | |
| Western Cape | Urban | City of Cape Town | 5 | 2 | | | 3 | 5 |
| | | Eden | | | 12 | 2 | | 1 |
| | Rural | Cape Winelands | | | 8 | 2 | 2 | 3 |
| | | West Coast | 1 | | 9 | | 2 | 2 |
| Eastern Cape | Urban | Buffalo City | 14 | | | | | |
| | | Nelson Mandela Bay | | | 10 | 3 | 1 | 1 |
| | Rural | OR Tambo | 8 | | 1 | 1 | 4 | 1 |
| | | Sarah Baartman | 7 | 2 | 2 | 2 | | 1 |
| Total (n=191): | | | 41 | 19 | 72 | 30 | 12 | 17 |

Numbers in cells refer to number of individual patient pathways observed for each organisational model within the district.
*iLembe* – undergoing rural to urban transition.

**Table 3** Characteristics of the study participants (n=191), within each organisational pattern, that may influence the degree of decentralisation achieved

| Participant characteristics | Category | Fan *n=41 | Chicken foot n=19 | Hub and spoke n=72 | Triangle n=30 | Line n=12 | Dot n=17 | Total n=191 |
|---|---|---|---|---|---|---|---|---|
| Age category (years) | 0–19 years | 3 (7) | 1 (5) | 3 (4) | 2 (7) | 0 | 0 | 9 (5) |
| | 20–59 years | 38 (93) | 18 (95) | 65 (90) | 27 (90) | 12 (100) | 17 (100) | 177 (92) |
| | ≥60 years | 0 | 0 | 4 (6) | 1 (3) | 0 | 0 | 5 (3) |
| Not receiving income | | 30 (73) | 14 (74) | 49 (68) | 18 (60) | 4 (33) | 9 (53) | 124 (65) |
| HIV positive | | 29 (71) | 16 (84) | 50 (70) | 18 (60) | 11 (92) | 7 (41) | 131 (69) |
| Baseline CD4<200 c/µL† | CD4<200 c/µL | 14 (48) | 6 (38) | 15 (30) | 5 (26) | 5 (45) | 2 (29) | 47 (35) |
| | CD4≥200 c/µL | 9 (31) | 5 (31) | 17 (34) | 6 (33) | 3 (27) | 3 (43) | 43 (33) |
| | Unreported CD4 | 6 (21) | 5 (31) | 18 (36) | 7 (39) | 3 (27) | 2 (29) | 41 (32) |
| Previous TB history | DS-TB any episode | 25 (61) | 14 (74) | 49 (68) | 19 (63) | 6 (50) | 8 (47) | 121 (63) |
| | MDR/RR-TB 1 episode | 3 (7) | 4 (21) | 6 (8) | 3 (10) | 3 (25) | 1 (6) | 20 (10) |
| | MDR/RR-TB≥2 episodes | 5 (12) | 1 (5) | 9 (13) | 1 (3) | 0 | 0 | 16 (9) |
| Baseline BMI<18.5 kg/m² | ≤18.5 kg/m² | 10 (24) | 7 (37) | 25 (35) | 7 (23) | 5 (42) | 4 (24) | 58 (49) |
| | ≥18.6 kg/m² | 19 (47) | 8 (42) | 26 (36) | 11 (34) | 0 (0) | 9 (52) | 74 |
| | Unreported BMI | 12 (29) | 4 (21) | 21 (29) | 12 (40) | 7 (58) | 4 (24) | 60 (51) |
| Comorbidities per participant (excluding HIV) | 1–3 comorbidities/pt | 4 (10) | 4 (21) | 15 (20) | 10 (33) | 2 (17) | 2 12) | 37 (19) |
| | >3 comorbidities/pt | 0 | 0 | 2 (3) | 1 (3) | 4 (33) | 2 (12) | 9 (5) |
| Second line TB drug resistance | MDR/RR-TB | 22 (53) | 11 (58) | 46 (64) | 22 (73) | 9 (76) | 12 (70) | 122 (64) |
| | Pre-XDR | 11 (27) | 6 (32) | 8 (11) | 4 (13) | 1 (8) | 1 (6) | 31 (16) |
| | XDR | 6 (15) | 1 (5) | 8 (11) | 2 (7) | 1 (8) | 1 (6) | 19 (10) |
| | Unknown resistance | 2 (5) | 1 (5) | 10 (14) | 2 (7) | 1 (8) | 3 (18) | 19 (10) |

*Number of patients.
†Within 3 months of diagnosis % refer to proportion of participants within each pattern.
BMI, body mass index; DS, drug-sensitive; MDR, multidrug-resistant ; pre-XDR, pre-extensively drug-resistant ; pt, participant; RR, rifampicin-resistant ; TB, tuberculosis.

treatment, low body mass index or higher number of comorbidities; however, the small sample size precludes robust statistical analysis.

### Implications of different patterns of organising care for people with MDR/RR-TB and the health system

We observed care pathways, reviewed patient records, visited healthcare facilities and interviewed facility staff to identify the implications of different organisational patterns on people with MDR/RR-TB and the health system (table 4). More decentralised models ('line' and 'dot') had advantages such as quick linkage to care, shorter hospital stays and less travel distance for diagnosis and treatment. However, these models of care were compromised at times through the risk of clinical practitioner shortages and less rigorous treatment monitoring. More centralised models ('fan' and 'chicken foot') had advantages such as multidisciplinary clinical, pharmacy, radiological and laboratory resources in one facility, but had disadvantages including long travel distances, treatment initiation delays and patient travel distance (table 1).

### DISCUSSION

This study used a novel approach, combining quantitative and qualitative methods and patient pathway analysis, to describe emerging patterns of decentralised care for MDR/RR-TB in South Africa. We identified six patterns of care organisation, ranging from highly centralised to highly decentralised models, with several different models operational in many study districts. The different models demonstrated distinct advantages and disadvantages for both people with MDR/RR-TB and the health system. Although exploratory, the patterns identified can contribute to developing a health systems typology for understanding how policy-driven models of service delivery are implemented in the context of variable resources and differing patient needs. Explicit recognition of the divergence of organisational models and cross-district learning around different models for organising care may provide opportunity for health system managers to optimise delivery of care in different contexts, and should be considered by policy developers when planning interventions to decentralise care for complex illness.

**Table 4** Advantages and challenges associated with each organisational care pattern

| Pattern name | Fan | Chicken foot | Hub and spoke | Triangle | Line | Dot |
|---|---|---|---|---|---|---|
| Advantages of the organisational model for people with MDR/RR-TB | Access to multidisciplinary team with experienced clinicians, drug availability. Access to acute tertiary (ICU, dialysis) and chronic tertiary care services (nephrology, cardiology, neurology). | People with less complex illness are able to be cared for in their districts. Ready referral pathway back to the centralised site for people with complex illness. | People are cared for in their districts, closer to home . Specialist TB hospitals may be better designed for long-term care than COEs. Staff are more familiar with individual people and their conditions and contexts, compared with COE. | People able to access care at different facilities, depending on preference. As a collective, the facilities have multidisciplinary resources. Expert conductor has the ability to navigate care and advocate for holistic care. Oversight for people with complex conditions. | Care close to home. Initial period of supervised inpatient care may provide nutrition support, side effect monitoring, linkage to financial grant support and patient education. | Care very close to home. Increased focus on community, patient-centred and family-centred care, with staff embedded within the community. Other decentralised services, for example, HIV care, family planning, are easily accessible. |
| Advantages of the organisational model for the health system | Logistically simple to compile statistics and monitor people with MDR/RR-TB and train specialised staff. Rapid response to change in diagnostic or drug regimens. Centralisation of expensive drug storage and control—minimise waste, expiry or shrinkage. | Reduces bed pressure on centralised sites, while maintaining strong clinical governance of people with complex illness. Less reliance on one facility per province (redundancy). | Strong connections between specialist TB hospital and clinics facilitates patient monitoring and surveillance. The district node taking a high degree of responsibility promotes accountability. Early treatment modification by TB specialists at the district level reduces admissions. | Oversight from an expert conductor facilitates rapid adjustment of patient care, reducing hospitalisation and optimising resource usage. Leverage of resources by senior clinicians. Flexibility around staffing allows system to buffer more easily. | Outpatient review within the community, with option for short hospital admission for treatment optimisation and stabilisation. Reduces patient transport (infection transmission, expense) and promotes family support, while minimising the need to provide services and drugs to every clinic. | Rapid diagnosis and treatment initiation. Reduced requirement for patient transport. Patient trust in local staff promotes adherence and treatment completion. Support from decentralised HIV programmes. |
| Challenges of the organisational model for people with MDR/RR-TB | Treatment delay, long admission times, hospital acquired infection, poor family centredness, loss of patient income. Community stigma of being at a COE for 'strong' TB. Disconnect from community chronic care services. | People with MDR/RR-TB need to adjust to care in different hospitals. Fragmentation of care. Travel between sites. Moderate to long admission times. | District sites may have reduced support from COEs. This may result in difficulty referring people with MDR/RR-TB requiring more complex care. Moderate to long admission times. | Care may be fragmented between the sites and people may become lost to follow-up between sites. Requires support from the province (travel, information technology services, training) which may not be consistent. | Sites may have reduced support from COEs. This may result in difficulty referring people requiring more complex care. Multidisciplinary team not always available. Complications may go undetected or there may be a delay in referring for higher levels of care. | Sites may have reduced support from COEs. This may result in difficulty referring people requiring more complex care. Multidisciplinary team less available. Complications may go undetected or delay in referral for higher-level care. |

**Table 4** Continued

| Pattern name | Fan | Chicken foot | Hub and spoke | Triangle | Line | Dot |
|---|---|---|---|---|---|---|
| Challenges of the organisational model for the health system | Reliance on a single centre—lack of redundancy. Skill set 'owned' by a few individuals at the COE's. Other HCWs see themselves as unable to care for MDR/RR-TB reducing willingness to care for these people. | Over-reliance on COE, with decentralised site poorly empowered to manage people with MDR/RR-TB. Providing safe patient transport. | Depending on the geographical size of the province, many people with MDR/RR-TB still travel far distances to visit the TB service or be admitted to hospital. | Reliance on retention of individual specialist physicians to maintain the programme. Tracking patient progress through the system. Providing safe transport. | Monitoring and supporting many MDR-TB sites. Providing sites with laboratory and IT infrastructure, drugs, trained staff and supervision. Ensuring quality of care provision and timely capturing of accurate data. | Monitoring and supporting many MDR-TB sites. Providing sites with laboratory and IT infrastructure, drugs, trained staff and supervision. Ensuring quality of care provision and timely capturing of accurate data. |

COE, Centre of Excellence ; HCWs, health care workers; IT, information technology; MDR, multidrug-resistant ; RR, rifampicin-resistant; TB, tuberculosis.

Stokes *et al* have previously described a framework for analysing models of care for patients with multi-morbidity.[31] A key component of this framework is the organisation of care across different levels of the health system. We and others have previously described other elements of this framework in relation to MDR/RR-TB in South Africa, including the theoretical basis for the decentralisation policy,[38] the clinical and social needs of the target population,[36 39] and the local adaptations of the decentralisation policy.[16 22 40] Our approach to identifying organisational models, which involved analysis of patient movement pathways, may be a useful tool and contribute valuable insights into understanding variation in policy implementation. This approach also provides important metrics, including the number of different facilities visited and travel distances. These metrics differed substantially between models in our study, with more centralised models associated with prolonged hospitalisation and multiple transfers between facilities. Although people with MDR/RR-TB who are treated with adequate doses of appropriate drugs rapidly become non-infectious,[41] there is increasing recognition that multiple patient transfers (often together with patients without TB) and prolonged hospitalisation may expose both people with MDR/RR-TB and others to unnecessary infectious risks, reduce retention in care and incur financial costs to people with MDR/RR-TB.[42 43] Studies in lower and middle income countries have shown that physical proximity of healthcare services can play an important role in the use of healthcare facilities.[44–46]

People with MDR/RR-TB in this cohort presented with multiple medical comorbidities, including mental illness and challenging social circumstances.[36] Our exploration of the advantages and disadvantages of the different organisational patterns highlights the trade-offs when having to consider people with complex or severe disease. Since it is not possible to decentralise all components of care for people with complex MDR/RR-TB, flexibility to provide differentiated or individualised care, which adapts to address the specific requirements of individuals or a subgroup of people with MDR/RR-TB, is particularly relevant.[20 22 47–49] For example, a flexible referral model like 'triangle', which uses centralised clinical oversight to direct patient care, or implementation of multiple healthcare models (as observed in the WC) may allow for a more seamless patient flow in response to the needs of people with MDR/RR-TB. Such flexible healthcare models, that can be organised in response to operational constraints and patient needs, may lead to contextually sensitive and optimal patient care that is safe, effective, person-centred, timely, efficient and equitable.[50–52] Although we did not collect data on patient costs, since centralised models were associated with longer hospital stays and travel distances, as well as fewer participants receiving an income, it is likely that patient costs were higher with centralised models. Fewer participants not receiving income fall into the line and dot patterns, which could point to these people with MDR/RR-TB being able

to keep on with their employment and livelihoods. The elimination of catastrophic costs to families affected by TB is one of the key pillars of the End-TB strategy.[53]

Care integration, and models that aim to overcome fragmentation between providers have been identified as important.[54] Ideally, a bio-psychosocial systems approach should be used, where the primary focus of healthcare is the client in the context of their family[55] and where adult and paediatric services are provided together in a single setting.[56] Reviews of family-centred care models in HIV care, indicate excellent adherence, retention in care and low mortality and/or loss to follow-up in both adult and child services.[57] Since HIV care in South Africa is highly decentralised, HIV status did not appear to affect the degree of decentralisation for people with MDR/RR-TB, even for people living with HIV who had low CD4 values. The dot model, where services are provided closest to the community is likely to be most family centred, but may be less ideal where home social circumstances are difficult, or where people with MDR/RR-TB require highly specialised care.[58] In the latter circumstances, models, such as triangle, which are intrinsically more flexible, and responsive to changing patient needs, while containing a mix of care expertise, levels and disciplines may be more appropriate.

Societal features, such as urbanisation, may have driven the emergence of certain models. However, we found no clear association between care patterns and epidemiological features of districts. For example, the most decentralised patterns (line and dot) were seen both in highly urbanised districts with large numbers of cases as well as in sparsely populated rural areas. These models appeared to be responses to differing pressures—the need to commence large numbers of people with MDR/RR-TB on treatment in urban areas, and the need to deliver care closer to people with MDR/RR-TB in remote areas. In rural districts in the EC and KZN, where several of the more decentralised models were available, these appeared to be driven by clinician champions, as we have previously described.[59] In the WC, the pre-existing strength of the primary healthcare system may have facilitated the preponderance of line and dot models.

The WHO annual TB report card on South Africa cautions that reported numbers of people diagnosed and commenced on TB treatment in 2020 had fallen by 25% compared with the same period in 2019 and that gains in the TB programme made over the past decade could be reversed by the COVID-19 pandemic.[60] Countries have been encouraged to mitigate COVID-19-related impacts on MDR/RR-TB services by expanding use of digital technologies for remote advice and support, improving community connectivity,[60] reducing the need for visits to healthcare facilities, encouraging home-based treatment and increasing use of strategies for facilitating patient drug supply. Lessons from the COVID-19 pandemic may offer lessons for organisation of MDR/RR-TB care delivery, in particular consideration of opportunities to improve connectivity to enhance decentralised care. We identified several examples where either telemedicine had been used, in line or dot models or physicians travelled to enable peripheral facilities to initiate and care for newly-diagnosed or more people with complex illness. In some instances, medication prescriptions or clinical records travelled in place of people with MDR/RR-TB. However, these adaptations were not a formal system response but rather an example of local adaptation, often with staff using their own professional networks, personal devices and internet connectivity for communication.[22] Description of patient movement patterns, critical appraisal of reasons for patient movement and identification of which travel might be avoided through other equally effective means, such as virtual consultations, may be a useful exercise for diseases that are complex, rare or require specialised care.[61–63]

Our study was undertaken prior to the widespread implementation of 6-month, all oral regimens for MDR/RR-TB (bedaquiline, pretomanid, linezolid and moxifloxacin), which may affect decentralisation models further. It is possible that these relatively simpler regimens may facilitate further decentralisation. However, this is countered by the need for ECG monitoring for cardiotoxicity, together with the need to provide ongoing training and mentorship to the staff involved. We observed that drug stockouts were a problem at decentralised sites—this may be a particular issue for more costly new drugs, due to concerns around shelf-life and wastage.

A limitation of our study is that it is not possible to clearly differentiate to what degree patient needs versus health system organisation and resources determined individual care pathways, but both are likely to have played a role. The substantial differences in care organisation, regardless of patient needs, reflect that there were multiple factors that were likely to determine care organisation. The sample size limited our ability to adequately assess care quality indicators, and so are unable to determine whether particular organisational models were associated with favourable patient outcomes. Further, we did not interview people with MDR/RR-TB, and so cannot draw conclusions about patient preferences. People with MDR/RR-TB who did not link to care, who died shortly after linking to care and those linking through the private sector were not included in this study. Since the entry point to the study was a microbiological diagnosis, we are unable to quantify the effect of decentralised management on diagnostic delay. We were unable to determine changes to the degree of decentralisation and evolution of movement patterns during the study period.

In this analysis, we also did not specifically interrogate the reasons for the emergence of the different organisational models. Models of care are often based on what was in place historically, and over time become less responsive to the changing needs of the person with MDR/RR-TB and to health system constraints.[64] Models of care are also often developed to bridge service delivery gaps rather than as a planned strategic response to an identified local need.[65 66] The development of models of care

is commonly an iterative process,[67 68] shaped by socio-political, economic, cultural, environmental and legal drivers.[69 70]

## CONCLUSION

With the challenge of striving towards the WHO targets of zero deaths, disease and suffering due to TB by 2035,[71] stronger and more flexible health systems for decentralising patient management of MDR/RR-TB will be required to accommodate varying treatment needs. Indeed, a key finding of our study is that flexible models for offering decentralised care have emerged in different settings, in response to differing patient needs. The study of patient care pathways is a useful approach to providing a health systems typology of the ways in which care is organised for complex conditions. Having more clearly defined and articulated models of care may assist policy planners and implementers to identify which organisational models are most responsive to patient needs, quality indicators and system constraints. Increased use of digital technologies should be explored to democratise specialised and individualised care for MDR/RR-TB and other complex illnesses.

**Author affiliations**
¹Division of Medical Microbiology, Department of Pathology, University of Cape Town, Cape Town, South Africa
²TB Centre, London School of Hygiene &Tropical Medicine, London, UK
³School of Public Health, University of the Western Cape, Bellville, South Africa
⁴Institute for Infectious Disease and Molecular Medicine and Wellcome Centre for Infectious Disease Research, University of Cape Town, Cape Town, South Africa
⁵School of Laboratory Medicine and Medical Sciences, University of KwaZulu-Natal College of Health Sciences, Durban, KwaZulu-Natal, South Africa
⁶Division of Infectious Diseases, University of Cape Town, Cape Town, South Africa
⁷South African Medical Research Council, Durban, South Africa
⁸Africa Health Research Institute, Somkhele, South Africa
⁹Institute of Tropical Medicine, Antwerp, Belgium
¹⁰Institute for Global Health and Development, Queen Margaret University, Edinburgh, UK
¹¹National Tuberculosis Control Programme, National Department of Health, Pretoria, South Africa
¹²School of Nursing and Public Health, University of KwaZulu-Natal College of Health Sciences, Durban, KwaZulu-Natal, South Africa
¹³Marshall Centre for Infectious DIsease Research and Training, School of Biomedical Sciences, University of Western Australia, Perth, Western Australia, Australia

**Acknowledgements** The authors wish to thank the Departments of Health of the Western Cape, Eastern Cape, KwaZulu-Natal, and acknowledge the staff at the NHLS for their tremendous input and assistance. We give special mention to the late Dr Iqbal Masters and Mrs Anna Maria Evans for their contributions to the study. We also appreciate the support of Staff Nurse Cheryl Liedeman and Dr Widaad Zemanay.

**Contributors** Conception and design of study: MN, HC, MM, AG, LD and KK. Planning of the work: MN, HC, MM, AG, LD, KK, KM, JB, ML, NN and WJ. Interrogation and analysis of laboratory reports and geo-mapping: JH, LD, SRLR, LM, MN and HC. Field visits and interviews: LD, SRLR, WJ, JH and LM. Analysis of field worker and interview data: LD, SRLR, WJ, JH, LM, KK, HC and MN. Conceptualising manuscript: LD and MN. Drafting manuscript and revisions: LD and MN. Reviewing manuscript and approval of final version: MN, HC, MM, AG, LD, KK, KM, JB, ML, NN, JH, SRLR, LM and WJ. MN is guarantor of the study.

**Funding** This study was supported by a Health Systems Research Initiative award from the Medical Research Council of the United Kingdom and the Wellcome Trust (MR/N015924/1). This UK funded award is part of the EDCTP2 programme supported by the European Union. HC is supported by a Wellcome Trust Fellowship. MN is supported by an Australian National Health and Medical Research Council Award (APP1174455). No conflicts of interest were reported.

**Map disclaimer** The inclusion of any map (including the depiction of any boundaries therein), or of any geographical or locational reference, does not imply the expression of any opinion whatsoever on the part of BMJ concerning the legal status of any country, territory, jurisdiction or area or of its authorities. Any such expression remains solely that of the relevant source and is not endorsed by BMJ. Maps are provided without any warranty of any kind, either express or implied.

**Competing interests** None declared.

**Patient and public involvement** Patients and/or the public were not involved in the design, or conduct, or reporting, or dissemination plans of this research.

**Patient consent for publication** Not applicable.

**Ethics approval** This study involves human participants and was approved by University of Cape Town (350/2016) London School of Hygiene and Tropical Medicine (11680). Waiver of informed consent was obtained from ethics committees.

**Provenance and peer review** Not commissioned; externally peer reviewed.

**Data availability statement** Data are available upon reasonable request. Data are available on reasonable request from the corresponding author.

**ORCID iDs**
Sacha Roxanne Le Roux http://orcid.org/0000-0002-0020-1414
Helen Cox http://orcid.org/0000-0002-6538-7192
Marian Loveday http://orcid.org/0000-0001-9205-9314
Norbert Ndjeka http://orcid.org/0000-0001-6200-1592
Mark Nicol http://orcid.org/0000-0002-1366-4805

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
