## [Reviewer comments · BMJ Open]

ARTICLE DETAILS

TITLE (PROVISIONAL)	The organisation of care for people receiving drug-resistant tuberculosis treatment in South Africa: a mixed methods study
AUTHORS	Dickson, Lindy; Le Roux, Sacha; Mitrani, Leila; Hill, Jeremy; Jassat, Waasila; Cox, Helen; Mlisana, Koleka; Black, John; Loveday, Marian; Grant, Alison; Kielmann, Karina; Ndjeka, Norbert; Moshabela, Mosa; Nicol, Mark

VERSION 1 – REVIEW

REVIEWER	Espinosa-Pereiro, Juan Vall d'Hebron University Hospital
REVIEW RETURNED	24-Apr-2023
GENERAL COMMENTS	GENERAL Although in this case the study focuses on the interaction between people with MDR-TB and the health care system, therefore in their role as patients, the use of this denomination in TB research is being abandoned as it suggests a 'passive' role of the ones included in the studies. Consider to minimise the use of 'patient' and consider, where feasible, alternatives such as 'affected by TB', 'people with TB', etc (as you did in line 411, for example). ABSTRACT The methods section in the abstract points out that patients were interviewed to check the movement pathways, but the main text says that this was not performed. Please clarify. BACKGROUND No comments. METHODS Consider elaborating somewhat more in this section about the overarching project: ecthics committee registration, leadership, and funding. Please clarify if the study period corresponds to the main project or to the substudy. Please include the definitions of preXDR and XDR used in this study, as it is mentioned in results (line 270), and the definitions have changed after the study period. Perhaps describe how the system was organised before de decentralisation and how the samples are managed in a programmatic way. From the text one can guess that samples are centralised in the National Health Laboratory RESULTS

	Would be important to assess whether the high rates of unemployment interact with patient mobilisation and whether they are due to the TB disease process. Line 287: 5 participants had a relative who died from TB. $5/194 = 2.5\%$, not 5%. Line 288: please specify the difference between ongoing treatment and ongoing disease. Line 289: Please specify if the 44 deaths are amongst the 139 alive at 6 months or overall. In the second case, it is not consistent with the other numbers provided as $194-139 = 55$, leaving 11 missing deaths. The authors state in the discussion that they could not correlate organisational models and patient outcomes. However, as results are provided in a 2-year period, can the authors provide if there was any association between treatment completion or adherence level and the movement pattern? This would relate to discussion lines 389 and 390. DISCUSSION Some statements are somewhat repetitive (e.g. lines 398-402, and 423-425), please review. In the paragraph starting in line 404, authors discuss the challenges that socioeconomic factors and comorbidities imply for MDR-TB care. Interestingly, one of the results is that movement pattern seems not to be affected by HIV status (even considering those $<200CD4?$). Consider to discuss this apparent contradiction that is quite relevant for MDR-TB. Is HIV care also decentralised? Limitations: Only four districts, so people moving out of these areas are not captured. A limitation is that the effect of decentralised management on diagnostic delay cannot be evaluated. Again, there is no information about the degree of adoption of decentralised care and thus evolution of movement patterns during the study period in the different districts. The study did not interview health system users, therefore there is no information about costs. Even without specific information, consider to add that centralised models imply longer hospital stays and longer travels, both related with increasing patient's costs, and hence catastrophic costs which elimination is another pillar of the End-TB strategy. For instance, in table 3, the fact that less participants not receiving income fall into the line and dot patterns could point that these patients can keep with their live. CONCLUSIONS One of the most interesting findings, from my point of view, is that different movement patterns were not dependent on the area but rather on patient's needs. This goes in line with more patientcentric and flexible care and against a 'one size fits all' from the programmatic perspective. Acknowledgements and funding Finding paragraph is included twice
REVIEWER	Ryckman, Theresa Johns Hopkins University Bloomberg School of Public Health

REVIEW RETURNED	23-Jun-2023
-----------------	-------------

GENERAL COMMENTS	This study used a combination of quantitative analysis from patient record review and qualitative analysis from healthcare worker interviews to describe different trajectories of patients receiving treatment for MDR/RR-TB in South Africa. I have a few major comments on the analysis and framing of results, and several more minor suggestions to improve clarity. Major points:  • Data is presented on patient/treatment outcomes, so it is not clear why the study does not include an assessment of the association between patient outcomes and the different trajectory categories. Including this evidence would strengthen findings. • It would be useful to discuss the findings and care models in the context of improved MDR regimens like BPaLM that are of shorter duration and may be less complex to administer and less burdensome to patients. • Figure 2: a caption explaining the figure would be helpful. For example, what do the different colors of the arrows represent? Is each panel showing one representative patient's care pathway? What do the asterisks in panel D represent? Are the panels in the same order as the order in which the trajectories are presented in the text? Minor points:  • The abstract states that patient interviews were done but strengths/limitations states they were not and they are not mentioned in the methods – this requires clarification. • Were there n=2649 total MDR/RR-TB patients in NHLS records from July-September 2016, of which only a subset (n=?) were from the study districts, and only a subset of those (n=195) were randomly sampled for this study? Was random sampling stratified by district? These sampling details are unclear as currently written. • Similarly, it is unclear how the subset of patients whose trajectories were analyzed was selected. • First paragraph of the results – references should be included for the statements being made about the 3 provinces. The authors could also consider moving this paragraph to the supplement, since it does not really contain results of the present study. • Lines 287-289: did the 31% of patients that had completed treatment not achieve cure, or is there no evidence on whether they achieved cure? • Lines 287-289: why don't the numbers add up to 194? • Table 2: including column totals at the bottom of the table would be helpful. • Table 3: difficult to interpret since categories are missing for some variables (e.g., ages 20-60, non-XDR, baseline GMI > 18.5 kg/m2, etc.)
--

VERSION 1 – AUTHOR RESPONSE

Reviewer: 1

Dr. Juan Espinosa-Pereiro, Vall d'Hebron University Hospital

		Reviewer comment	Response / changes
1	General	Although in this case the study focuses on the interaction between	Thank you for this suggestion, we have amended as suggested.
		people with MDR-TB and the health care system, therefore in their role as patients, the use of this denomination in TB research is being abandoned as it suggests a 'passive' role of the ones included in the studies. Consider to minimise the use of 'patient' and consider, where feasible, alternatives such as 'affected by TB', 'people with TB', etc (as you did in line 411, for example).	
2	Abstract	The methods section in the abstract points out that patients were interviewed to check the movement pathways, but the main text says that this was not performed. Please clarify.	Our apologies, we have reported on patient interviews elsewhere, and this is not included. We have amended this.
3	Methods	Consider elaborating somewhat more in this section about the overarching project: ethics committee registration, leadership, and funding.	We have added the details requested as well as included an additional Supplementary Figure A, to show how this study is nested within the overarching project.
		Please clarify if the study period corresponds to the main project or to the sub study.	The study periods are the same – we have clarified this.

		Please include the definitions of pre-XDR and XDR used in this study, as it is mentioned in results (line 270), and the definitions have changed after the study period.	We have added these details as supplementary information (Table S2).
		Perhaps describe how the system was organised before decentralisation and how the samples are managed in a programmatic way. From the text	We have added details explaining how prior to decentralization, people with MDR/RR-TB in South Africa were primarily managed within facilities specializing in

		one can guess that samples are centralised in the National Health Laboratory	MDR/RR-TB care, frequently with long hospital stays before being discharged under the ongoing care of the specialized facilities. We have also included details of how all TB samples in the state sector in South Africa are sent to laboratories which form part of the National Health Laboratory Services (NHLS) for testing. The NHLS has a centralized data repository which records all samples sent to laboratories within the network, together with results and patient details.
4	Results	Would be important to assess whether the high rates of unemployment interact with patient mobilisation and whether they are due to the TB disease process.	We agree that these factors interact. Unfortunately, we do not have the data sources to clarify the primary drivers of unemployment or mobilization in this cohort.
		Line 287: 5 participants had a relative who died from TB. $5/194 = 2.5\%$, not 5%.	Apologies, corrected to $10/194 = 5\%$

		Line 288: please specify the difference between ongoing treatment and ongoing disease.	We have clarified that 'ongoing disease' referred to one patient who had repeatedly failed treatment and had been referred for palliative care.
		Line 289: Please specify if the 44 deaths are amongst the 139 alive at 6 months or overall. In the second case, it is not consistent with the other numbers provided as $194-139 = 55$, leaving 11 missing deaths.	We apologize for the lack of clarity regarding denominators in this section. We have simplified this reporting and now report that of the 191 participants 80% (n=153) were alive at 6 months, and 77% (n=148) were alive at 2 years after laboratory diagnosis.
		The authors state in the discussion that they could not correlate	Thank you for this suggestion. We were hesitant to draw conclusions

		organisational models and patient outcomes. However, as results are provided in a 2-year period, can the authors provide if there was any association between treatment completion or adherence level and the movement pattern? This would relate to discussion lines 389 and 390.	from associations between movement pattern and outcomes, given the relatively small sample size. However, since this is of interest, we have added these outcomes to Table 1, and added text to the results and discussion to reflect on these findings.
5	Discussion	Some statements are somewhat repetitive (e.g. lines 398-402, and 423-425), please review.	We agree and have edited the discussion substantially to reduce redundancy.

		In the paragraph starting in line 404, authors discuss the challenges that socioeconomic factors and comorbidities imply for MDR-TB care. Interestingly, one of the results is that movement pattern seems not to be affected by HIV status (even considering those <200CD4?). Consider to discuss this apparent contradiction that is quite relevant for MDR-TB. Is HIV care also decentralised?	This is an important observation. Since HIV care in South Africa is highly decentralised, HIV status did not appear to affect the degree of decentralisation for people with MDR/RR-TB, even for people living with HIV who had low CD4 values. We have added these details.
6	Limitations	Only four districts, so people moving out of these areas are not captured.	We have captured all movements of study participants, even when this occurred outside of the study districts. We have noted this in the methods section.
		A limitation is that the effect of decentralised management on diagnostic delay cannot be evaluated.	This is correct, we have added this to the limitation section.
		Again, there is no information about the degree of adoption of decentralised care and thus evolution of movement patterns	We were unable to determine changes to the degree of decentralization and evolution of movement patterns during the
		during the study period in the different districts.	study period. We have added this limitation.

		The study did not interview health system users, therefore there is no information about costs. Even without specific information, consider to add that centralised models imply longer hospital stays and longer travels, both related with increasing patient's costs, and hence catastrophic costs which elimination is another pillar of the End-TB strategy. For instance, in table 3, the fact that less participants not receiving income fall into the line and dot patterns could point that these patients can keep on with their lives.	Thank you for this useful suggestion. We have added the following text to the discussion: "Although we did not collect data on patient costs, since centralized models were associated with longer hospital stays and travel distances, as well as fewer participants receiving an income, it is likely that patient costs were higher with centralized models. The elimination of catastrophic costs to families affected by TB is one of the key pillars of the End-TB strategy."
7	Conclusion	One of the most interesting findings, from my point of view, is that different movement patterns were not dependent on the area but rather on patient's needs. This goes in line with more patientcentric and flexible care and again a 'one size fits all' from the programmatic perspective.	Thank you for this comment. We agree and have added a sentence to reflect this in the conclusion.
8	Acknowledgements and funding	Funding paragraph is included twice.	Corrected.

Reviewer: 2

Dr. Theresa Ryckman, Johns Hopkins University Bloomberg School of Public Health

	Reviewer comment	Response / changes
--	------------------	--------------------

Major points	Data is presented on patient/treatment outcomes, so it is not clear why the study does not include an assessment of the association between patient outcomes and the different trajectory categories. Including this evidence would strengthen findings.	Thank you for this suggestion, which was also made by reviewer 1. We were hesitant to draw conclusions from associations between movement pattern and outcomes, given the relatively small sample size. However, since this is of interest, we have added these outcomes to Table 1, and added text to the results and discussion to reflect on these findings
	It would be useful to discuss the findings and care models in the context of improved MDR regimens like BPaLM that are of shorter duration and may be less complex to administer and less burdensome to patients.	We agree and have added text to the discussion: “Our study was done prior to the widespread implementation of 6month, all oral regimens for MDR/RR-TB (bedaquiline, pretomanid, linezolid and moxifloxacin [BPaLM]), which may affect decentralization models further. It is possible that these relatively simpler regimens may facilitate further decentralization. However, this is countered by the need for ECG monitoring for cardiotoxicity. We observed that drug stockouts were a problem at decentralized sites – this may be a particular issue for more costly new drugs, due to concerns around shelf-life and wastage.”
	Figure 2: a caption explaining the figure would be helpful. For example, what do the different colours of the arrows represent? Is each panel showing one representative patient’s care pathway? What do the asterisks in panel D represent? Are the panels in the same order as the order in which the trajectories are presented in the text?	Apologies for the lack of explanation. We have added a caption explaining the figure. Each coloured line depicts a single participant travelling in a pathway between the specific levels of health care facilities found in that model. Arrows represent direction of participant movement direction between health care facilities. Stars represent participants who only visited a single facility across their diagnostic and treatment pathway.

Minor points	The abstract states that patient interviews were done but strengths/limitations state they were not and they are not mentioned in	Our apologies, we have reported on patient interviews elsewhere, and this is not included. We have amended this.
	the methods – this requires clarification.	
	Was there n=2649 total MDR/RR-TB patients in NHLS records from JulySeptember 2016, of which only a subset (n=?) were from the study districts, and only a subset of those (n=195) were randomly sampled for this study? Was random sampling stratified by district? These sampling details are unclear as currently written.	Your interpretation is correct; we have amended the text to clarify.
	Similarly, it is unclear how the subset of patients whose trajectories were analysed was selected.	We have clarified that all 194 correctly diagnosed participants, who were randomly selected, were included in the trajectory analysis, with 3 patients with insufficient data points to create a trajectory.
	First paragraph of the results – references should be included for the statements being made about the 3 provinces.	We have added references to the table summarizing these statistics.
	The authors could also consider moving this paragraph to the supplement, since it does not really contain results of the present study.	We agree and have removed this paragraph and referred the reader to the summary Table S1.
	Lines 287-289: did the 31% of patients that had completed treatment not achieve cure, or is there no evidence on whether they achieved cure?	We have added Table S2 to the supplementary text to clarify definitions – treatment completion refers to participants who completed treatment without evidence of sustained negative cultures, versus treatment cure which requires cultureconfirmation.
	Lines 287-289: why don't the numbers add up to 194?	Apologies for this – various denominators were used (191 vs 194) at different points. We have amended this section to clarify denominators.

	Table 2: including column totals at the bottom of the table would be helpful.	We have added the totals to this table.
	Table 3: difficult to interpret since categories are missing for some variables (e.g., ages	We have included these additional categories.
	20-60, nonkg/m2, line BMI > 18.5 etc.)	

VERSION 2 – REVIEW

REVIEWER	Ryckman, Theresa Johns Hopkins University Bloomberg School of Public Health
REVIEW RETURNED	13-Sep-2023

GENERAL COMMENTS	I would like to thank the authors for responding to my suggestions and incorporating corresponding edits to their study. I'm still a little confused about the definitions of different treatment outcomes – particularly completion vs. cure. In Table S2, it is stated that “completed treatment” means there was no evidence of sustained negative cultures (e.g., the “completed treatment” and “cured” categories appear to be mutually exclusive). However, in Table 1, for instance, only the “completed treatment” category is listed – I think because the cures are included here (maybe?) – but based on Table S2 they should be broken out? In Figure 1, it would be helpful to include a description of what the different health care facilities represent (green cross vs. red cross vs. blue H vs. orange H). Other than these minor suggestions the authors have adequately responded to my feedback and I have no further comments.
---

VERSION 2 – AUTHOR RESPONSE

We apologize for the lack of clarity regarding definitions of treatment outcomes. Reviewer 2 is correct that, in Table 1, the "completed treatment" category includes both patients who have completed treatment as well as those that have achieved "cure". We have amended the table to reflect this. Since numbers are small, for simplicity we have not further disaggregated these into the two categories of "completed treatment" and "cure".

We have added details of the categories of health care facility represented by the different symbols to the legend for Figure 1. Apologies for omitting these.

We have edited the abstract in line with authors instructions and included a data sharing statement at the end of the manuscript.